# Reference rate for post-tonsillectomy haemorrhage in Australia—A 2000–2020 national hospital morbidity database analysis

**Jonathan C. Li** [1]*, **Martin Forer**[1], **David Veivers** [1,2]

**1** Department of Otolaryngology Head and Neck Surgery, Royal North Shore Hospital, Sydney, NSW, Australia, **2** Northern Clinical School, The University of Sydney, Sydney, NSW, Australia

* jcyli@icloud.com

## Abstract

This study aims to provide a national benchmark rate of post-tonsillectomy haemorrhage (PTH) in Australia. Using data from Australia's National Hospital Morbidity Database (NHMD) from 1 July 2000 to 30 June 2020, we have conducted a nation-wide population-based study to estimate a reference rate of PTH. Outcomes of interest included the overall rate and time-trend of PTH, the relationship between PTH rates with age and gender as well as the epidemiology of tonsillectomy procedures. A total of 941,557 tonsillectomy procedures and 15,391 PTH episodes were recorded for the study period. Whilst the incidence of tonsillectomy procedures and the number of day-stay tonsillectomy procedures have increased substantially over time, the overall rate of PTH for all ages has remained relatively constant (1.6% [95% CI: 1.61 to 1.66]) with no significant association observed between the annual rates of PTH and time (year) (Spearman correlation coefficient, $R_s$ = 0.24 (95% CI: -0.22 to 0.61), $P$ = 0.3). However, the rate of PTH in adults (aged 15 years and over) experienced a statistically significant mild to moderate upward association with time (year) $R_s$ = 0.64 (95% CI: 0.28 to 0.84), $P$ = 0.003. Analysis of the odds of PTH using the risk factors of increasing age and male gender showed a unique age and gender risk pattern for PTH where males aged 20 to 24 years had the highest risk of PTH odds ratio 7.3 (95% CI: 6.7 to 7.8) compared to patients aged 1 to 4 years. Clinicians should be mindful of the greater risk of PTH in male adolescents and young adults. The NHMD datasets can be continually used to evaluate the benchmark PTH rate in Australia and to facilitate tonsillectomy surgical audit activities and quality improvement programs on a national basis.

## Introduction

Tonsillectomy is one of the most performed surgical procedures in the field of otolaryngology. In Australia, over 50,000 tonsillectomies are performed annually [1]. The common indications for tonsillectomy include obstructive sleep disordered breathing and recurrent acute tonsillitis. Depending on the clinical context, tonsillectomy is carried out either as a sole procedure or in combination with other procedures. Tonsillar haemorrhage following tonsillectomy is a significant post-operative complication with the potential to cause serious morbidity and death [2, 3]. Patients with post-tonsillar haemorrhage (PTH) may require hospital re-admission

**Data Availability Statement:** The datasets analysed for the current study are in the public domain and are available from the Australian Institute of Health and Welfare website repository

(https://www.aihw.gov.au/reports/hospitals/procedures-data-cubes/contents/data-cubes).

**Funding:** The author(s) received no specific funding for this work.

**Competing interests:** The authors have declared that no competing interests exist.

including the urgent return to theatre under general anaesthesia for operative haemostasis. Due to the sheer numbers of tonsillectomies that are performed, even seemingly low rates of PTH at the population level can affect a large number of patients and generate a significant cost burden to the healthcare system and to our society.

Surgical audits and quality assurance activities are an essential part of surgical practice to improve patient safety. Contemporary clinical practice guidelines on tonsillectomy recommend clinicians to continually self-monitor their rates of PTH and compare their personal PTH rates with those from published reports and national benchmarks [4]. However, there has not been a population-based study or large multi-centre study conducted in Australia to provide a reference rate for PTH.

Australian PTH rates that are currently available have been derived from single-centre studies, often containing limited study population sizes [5–11]. As a result, these studies may be subject to inherent biases in the reporting of factors that have been shown to be determinants of PTH including age, gender, surgical technique, surgeon's level of experience and underlying indication for tonsillectomy [12]. Furthermore, the reporting of PTH rates vary significantly in the literature, making it difficult for comparisons. This variability is largely owed to the different definitions of PTH that are used by studies. PTH can be divided into primary ($< 24$ hours) and secondary ($> 24$ hours) based on the timing of onset following the index operation. Studies that define PTH by episodes that require re-operation tend to have lower PTH rates than studies that use a definition based on the patient's self-reporting of bleeding or on the number of emergency room or hospital admissions [13]. Accordingly, a national benchmark rate for PTH that is transparently derived and representative of the practice of tonsillectomy across Australia can facilitate quality improvement in tonsillectomy on a national scale. Given that age and gender are significant predictors of PTH [12, 14–16], a benchmark rate of PTH should be adjusted for these two universal risk factors.

The primary aim of this study is to devise an Australian PTH reference rate and to evaluate its time-trend using publicly available data from the National Hospital Morbidity Database (NHMD). The estimated reference rate for PTH will be based on the return to theatre rate, specific for age and gender and devised using population-based data over the time period 2000 to 2020. A secondary aim of this study is to evaluate the epidemiology of tonsillectomy procedures to provide the context for the interpretation of the PTH rates in Australia.

## Materials and methods

### Data sources

The study utilises the procedure datasets contained within the NHMD which is a comprehensive collection of clinical and non-clinical records associated with each episode of admitted patient care within Australia. The NHMD commenced in 1993–94 and is maintained and governed by the Australian Institute of Health and Welfare (AIHW), an Australian Government's health information agency [17]. All state and territory health authorities under the National Health Information Agreement (NHIA) are required to report to the NHMD annually following each Australian financial year. The coverage of the NHMD includes almost all public and private hospitals and free-standing day hospital facilities in Australia [17].

The annual hospital procedure datasets from 2000–01 through to 2019–20 (total period coverage from 1 July 2000 to 30 June 2020) were retrieved for this study. This data is available in the public domain and is freely accessible from AIHW's website [1]. The data is de-identified and contains the procedure counts categorised by procedures codes. These codes are classified according to the Australian Classification of Health Interventions (ACHI) which is an Australian coding standard based on the Medicare Benefits Schedule (MBS). The variables

contained within these datasets are age (grouped in 5-year intervals), gender, and hospital care type (i.e., day-stay versus overnight care).

The source population for incidence rate calculations was the annual mid-year resident population compiled and estimated from the Census population by the Australian Bureau of Statistics (ABS) [18].

## Ethical consideration

This study was approved by the Northern Sydney Local Health District Human Research Ethics Committee (2022/ETH00229).

## Definition of outcome measures

For our study, we have used the definition of PTH as episodes that were clinically severe to require the return to theatre for surgical haemostasis (ACHI code 41797–00 *Arrest of haemorrhage following tonsillectomy*). The count of this procedure code is synonymous with the return to theatre rate for haemorrhage control following tonsillectomy. Similarly, patients who had undergone a tonsillectomy procedure were identified using the following ACHI codes; *Tonsillectomy with adenoidectomy* (41789–01), *tonsillectomy without adenoidectomy* (41789–00) and *tonsillectomy with uvulopharyngopalatoplasty (UPPP)* (41786–01). The PTH rate was calculated by dividing the number of return to theatre episodes (41797–00) by the total number of tonsillectomy procedures (41789–01 + 41789–00 + 41786–01). The annual incidence for tonsillectomy procedures was calculated by dividing the yearly total number of tonsillectomy procedures by the mid-year resident population. For analyses involving the dichotomous comparison between adults and children, the age groups containing ages 1 to 14 were collectively defined as the paediatric age group and the age groups aged 15 years and above defined as the adult age group.

## Statistical analysis

Categorical variables were analysed using descriptive statistics to determine frequency of counts, percentages of total, means and standard deviation. The 95% confidence intervals (95% CI) for rate calculations were estimated using the Poisson regression model. Time-trend analysis of annual PTH rates was performed using the Spearman's rank correlation coefficient ($R_s$) to examine the strength of the association between the annual PTH rate (the dependent variable) and year (independent variable). The Pearson chi-square test was used where appropriate, and binomial logistic regression was used to analyse the univariate associations between the PTH rate with age and gender and are presented as odd ratios with 95% CI. All statistical tests were two sided with significance set at $P < 0.05$. Study data was collated using Excel (Microsoft Corporation) and statistical analyses were performed using StatsDirect 3.3.5 (StatsDirect Ltd., Wirral, UK).

## Reporting guideline

The reporting of this study conforms to the STROBE statement.

## Results

### Post-tonsillectomy haemorrhages in Australia

**Overview.** A total of 15,391 episodes of PTH and 941,557 tonsillectomy procedures were recorded between 1 July 2000 and 30 June 2020. This equates to an overall PTH rate of 1.63%

(95% CI: 1.61 to 1.66) for the study period. The average annual PTH rate for the study period was 1.6 ± 0.1% [mean ± standard deviation (SD)] (S1 Table).

**Time-trends of post-tonsillectomy rates.** Time series of the annual PTH rates is shown in Fig 1A. It demonstrates no discernible monotonic time-trend for the study period. Correspondingly, the Spearman's rank correlation showed no significant association between annual PTH rate and time (year) ($R_s$ = 0.24 [95% CI: -0.22 to 0.61], $P$ = 0.3).

Further analyses were performed separately on the annual PTH rates for the adult and the paediatric patient populations (Table 1) with their time-series shown in Fig 1B. The paediatric age group had an average annual PTH rate of 1.0 ± 0.1% (mean ± SD) for the study period. Similarly, the Spearman's correlation test showed no statistically significant association between the annual PTH rates and time (year) for the paediatric age group (Rs = 0.03 [95% CI: -0.42 to 0.47], P = 0.9). In contrast, a higher average annual PTH rate was calculated for the adult age group at 2.8 ± 0.3% (mean ± SD). The Spearman's correlation test for association between annual PTH rates and time (year) performed for the adult age group showed a statistically significant, mild to strongly positive correlation, indicating an increasing trend of adult PTH rates over time (year) (Rs = 0.64 [95% CI: 0.28 to 0.84], P = 0.003).

**Relationship of haemorrhage rates with age and gender.** Fig 2 shows the age-specific PTH rates calculated for the total study period for males and females. Two distinct gender patterns are evident. For both genders, the rates of PTH increased marginally with increasing age from age 1 to 14. From the age of 15, the increase was notably marked and sustained for the male gender with the rate of PTH reaching a maximum in the 20 to 24 years age group. The comparison of the odds of PTH between male and female were also calculated for each age group and are presented in Table 2. It demonstrates that male patients had a higher risk of PTH that was statistically significant as compared to females for all age groups except for patients aged 5 to 9.

For both male and female, patients aged 1 to 4 years had the lowest rate of PTH (0.8% and 0.7%, respectively). Males aged 20 to 24 years had over 7 times the odds of PTH than males aged 1 to 4 years (odds ratio 7.3 [95% CI: 6.7 to 7.8], P < 0.0001). Correspondingly, females in the 20 to 24 years age group had 3 times the odds of PTH than females in the 1 to 4 years age group (odds ratio 3.1 [95% CI: 2.8 to 3.4], P < 0.0001). Further comparison of the odds of PTH between age groups for each gender using the 1 to 4 years age group as the reference age group is presented in S2 Table.

The odds of PTH between the paediatric and adult age groups for both genders were also calculated (Table 3). It shows that, collectively, adult males were twice as likely to experience a PTH episode compared to adult females (odds ratio of 4.3 [95% CI: 4.1 to 4.9], P < 0.0001, versus 2.1 [95% CI: 2.0 to 2.2], P < 0.0001, respectively). Table 3 also shows that adult female patients had twice the odds of experiencing PTH compared to paediatric patients whereas adult male patients had 4 times the odds of experiencing a PTH episode compared to paediatric patients.

## Epidemiology of tonsillectomy in Australia

**Characteristics of tonsillectomy procedures.** Characteristics of 941,557 tonsillectomy procedures recorded during the study period are also shown in S3 Table. These tonsillectomy procedures consisted of tonsillectomy with adenoidectomy (61.2%), tonsillectomy without adenoidectomy (36.8%) and tonsillectomy with UPPP (2%). The gender ratio had a slight preponderance for females (51.3%). The young paediatrics age groups contributed the highest number of tonsillectomy procedures (age 1 to 4 years with 28.7% and age 5 to 9 years with 26.9%). The data also included counts of tonsillectomy performed on infants (age <1 year)

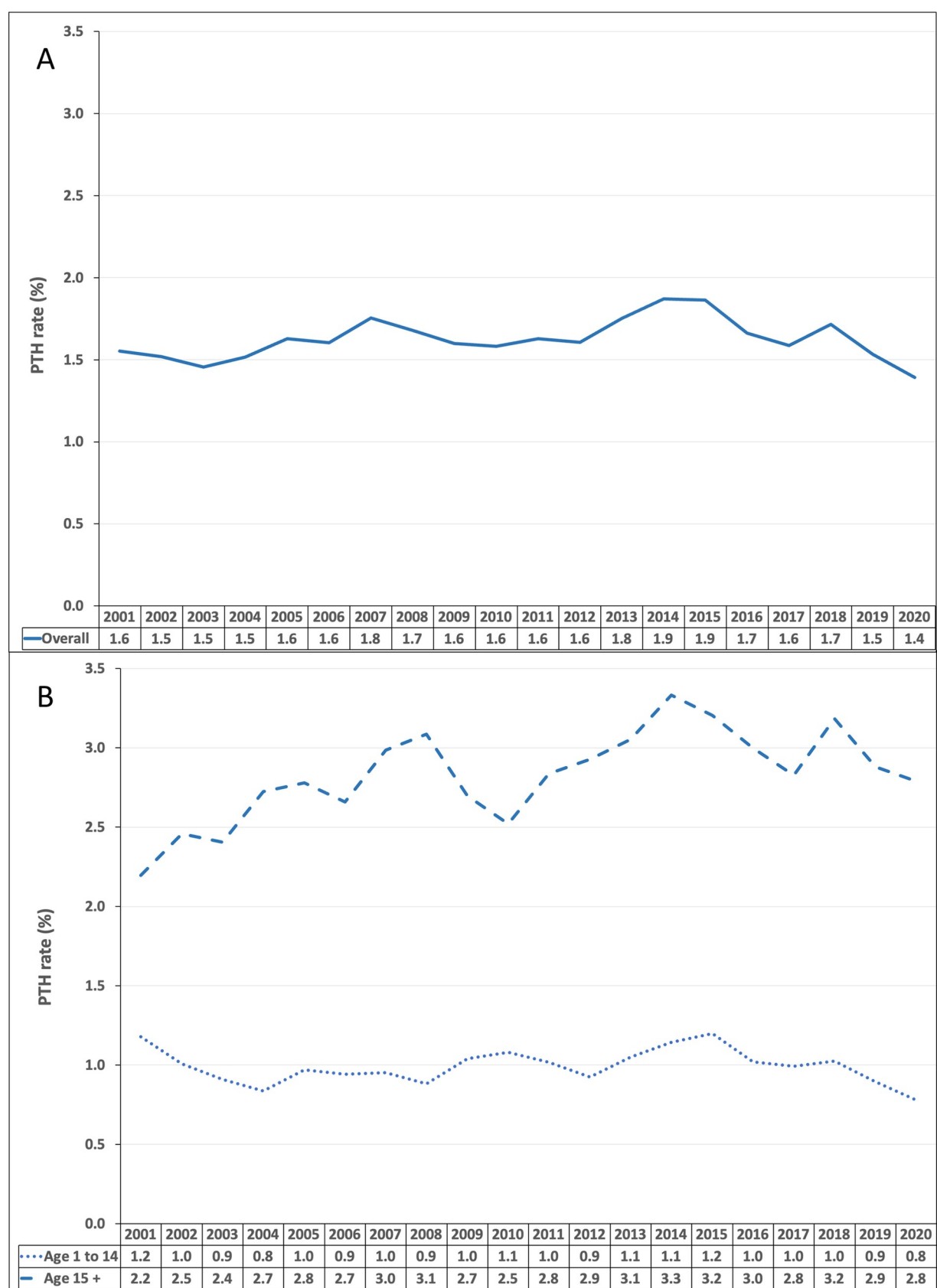

**Fig 1. Annual post-tonsillectomy haemorrhage rates, 2000–01 to 2019–20.** A; Overall annual rates of PTH calculated for all ages. The absolute values of PTH episodes and tonsillectomy procedures used to calculate the PTH rates are available in S1 Table. B; Annual rates of PTH calculated for the adult (age over 15 years) and paediatric age groups (age 1 to 14). PTH = post-tonsillectomy haemorrhage. Year represents the Australian financial year-ended period from 1 July to 30 June. Data obtained from the National Hospital Morbidity Database.

with an incidence rate of 0.04% (1 in every 2,500 tonsillectomy procedures). Analysis by treatment type showed that an overwhelming majority of tonsillectomy procedures were performed as an inpatient procedure (91.3%).

**Incidence of tonsillectomy procedures.**   Over the past two decades, there has been a sustained increase in the incidence rate of tonsillectomy procedures performed in Australia. The incidence rate rose from 156.5 per 100,000 persons (95% CI: 154.7 to 158.3) in 2000–01 to a peak of 264.4 per 100,000 persons (95% CI: 262.4 to 266.5) in 2016–17 (Fig 3 and S4 Table). From 2016–17 onwards, a plateauing and trending decrease in the incidence rate of tonsillectomy procedures was observed (Fig 3).

**Tonsillectomy time-trends by procedure types, age-specific incidence, and proportion of day-stay procedures.**   Contained within Fig 4 are other time-series plots related to tonsillectomy procedures in Australia. Fig 4A shows that tonsillectomy with adenoidectomy has consistently been the most performed tonsillectomy procedure type, with a prominent

**Table 1.  Annual rates of post-tonsillectomy haemorrhage in Australia for the adult and paediatric population, 2000–01 to 2019–20.**

| Year | Adult age group | | | | Paediatric age group | | | |
|------|-----|------------------------|--------------|-----------|-----|------------------------|--------------|-----------|
|      | PTH | Tonsillectomy procedures | PTH rate (%) | 95% CI | PTH | Tonsillectomy procedures | PTH rate (%) | 95% CI |
| **2000–01** | 241 | 10,974 | 2.2 | (1.93, 2.49) | 226 | 19,180 | 1.2 | (1.03, 1.34) |
| **2001–02** | 294 | 11,963 | 2.5 | (2.18, 2.76) | 220 | 21,849 | 1.0 | (0.88, 1.15) |
| **2002–03** | 290 | 12,062 | 2.4 | (2.14, 2.70) | 191 | 20,990 | 0.9 | (0.79, 1.05) |
| **2003–04** | 320 | 11,750 | 2.7 | (2.43, 3.04) | 174 | 20,821 | 0.8 | (0.72, 0.97) |
| **2004–05** | 341 | 12,267 | 2.8 | (2.49, 3.09) | 207 | 21,357 | 1.0 | (0.84, 1.11) |
| **2005–06** | 362 | 13,620 | 2.7 | (2.39, 2.95) | 204 | 21,669 | 0.9 | (0.82, 1.08) |
| **2006–07** | 432 | 14,471 | 3.0 | (2.71, 3.28) | 211 | 22,173 | 1.0 | (0.83, 1.09) |
| **2007–08** | 470 | 15,230 | 3.1 | (2.81, 3.38) | 237 | 26,855 | 0.9 | (0.77, 1.00) |
| **2008–09** | 432 | 16,029 | 2.7 | (2.45, 2.96) | 327 | 31,431 | 1.0 | (0.93, 1.16) |
| **2009–10** | 420 | 16,672 | 2.5 | (2.28, 2.77) | 334 | 30,943 | 1.1 | (0.97, 1.20) |
| **2010–11** | 476 | 16,784 | 2.8 | (2.59, 3.10) | 339 | 33,318 | 1.0 | (0.91, 1.13) |
| **2011–12** | 510 | 17,415 | 2.9 | (2.68, 3.19) | 311 | 33,661 | 0.9 | (0.82, 1.03) |
| **2012–13** | 568 | 18,598 | 3.1 | (2.81, 3.32) | 364 | 34,655 | 1.1 | (0.95, 1.16) |
| **2013–14** | 606 | 18,189 | 3.3 | (3.07, 3.61) | 417 | 36,449 | 1.1 | (1.04, 1.26) |
| **2014–15** | 599 | 18,694 | 3.2 | (2.95, 3.47) | 453 | 37,781 | 1.2 | (1.09, 1.31) |
| **2015–16** | 612 | 20,414 | 3.0 | (2.77, 3.25) | 432 | 42,374 | 1.0 | (0.93, 1.12) |
| **2016–17** | 592 | 20,946 | 2.8 | (2.60, 3.06) | 438 | 44,084 | 1.0 | (0.90, 1.09) |
| **2017–18** | 647 | 20,330 | 3.2 | (2.94, 3.44) | 443 | 43,169 | 1.0 | (0.93, 1.13) |
| **2018–19** | 562 | 19,525 | 2.9 | (2.65, 3.13) | 368 | 41,124 | 0.9 | (0.81, 0.99) |
| **2019–20** | 437 | 15,672 | 2.8 | (2.53, 3.06) | 278 | 35,677 | 0.8 | (0.69, 0.88) |
| **Mean** | 461 | 16,080 | 2.8 | - | 309 | 30,978 | 1.0 | - |
| **SD** | 124 | 3,200 | 0.3 | - | 96 | 8,513 | 0.1 | - |

Data obtained from the National Hospital Morbidity Database for period 1 July 2000 to 30 June 2020. PTH = post-tonsillectomy haemorrhage; CI = confidence interval; SD = standard deviation. Data are presented as absolute number of PTH and tonsillectomy procedures for the adult (age 15 years and over) and paediatric (age 1 to 14 years) patient population. PTH defined by re-operations for haemostasis. Annual PTH rates are calculated by dividing the number of PTH with the number of tonsillectomy procedures.

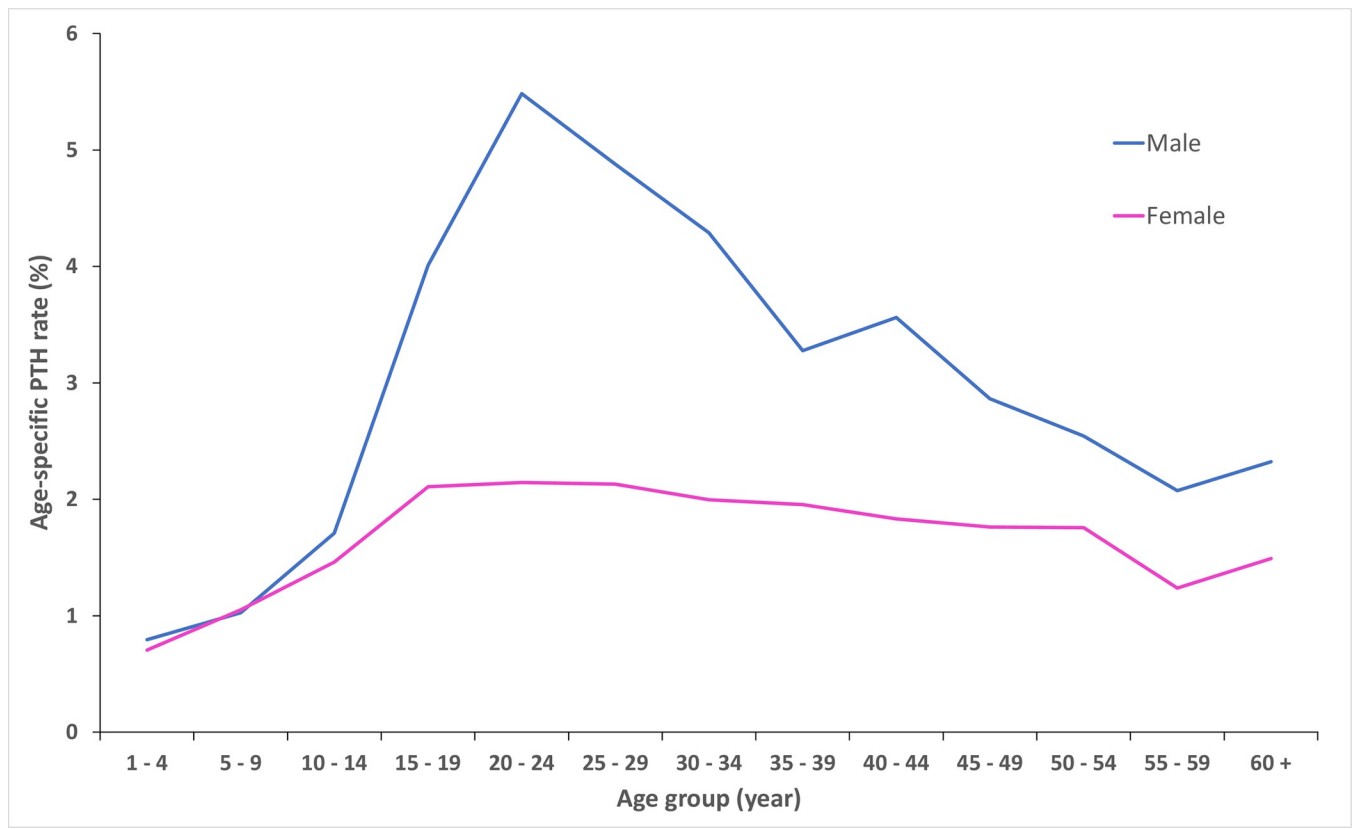

**Fig 2. Gender and age specific rates of post-tonsillectomy haemorrhage in Australia, 2000–01 to 2019–20.** Age-specific incidence rates of post-tonsillectomy haemorrhage (PTH) calculated by 5-year age groups for the male and female gender. Data obtained from the National Hospital Morbidity Database.

**Table 2. Comparison of age-specific post-tonsillectomy haemorrhage rates between the male and female gender.**

| | Female | | | Male | | | Comparison | | |
|---|---|---|---|---|---|---|---|---|---|
| Age | PTH | Tonsillectomy procedures | PTH rate (%) | PTH | Tonsillectomy procedures | PTH rate (%) | Odds ratio | 95%CI | P value |
| 1–4 | 766 | 108,477 | 0.7 | 1284 | 161,783 | 0.8 | 1.1 | (1.0, 1.2) | 0.01 |
| 5–9 | 1276 | 121,576 | 1.0 | 1352 | 131,971 | 1.0 | 1.0 | (0.9, 1.1) | 0.54 |
| 10–14 | 820 | 56,167 | 1.5 | 676 | 39,578 | 1.7 | 1.2 | (1.1, 1.3) | 0.003 |
| 15–19 | 1637 | 77,661 | 2.1 | 1325 | 33,045 | 4.0 | 1.9 | (1.8, 2.1) | * |
| 20–24 | 1034 | 48,184 | 2.1 | 1362 | 24,836 | 5.5 | 2.6 | (2.4, 2.9) | * |
| 25–29 | 502 | 23,567 | 2.1 | 741 | 15,193 | 4.9 | 2.4 | (2.1, 2.6) | * |
| 30–34 | 351 | 17,581 | 2.0 | 625 | 14,573 | 4.3 | 2.2 | (1.9, 2.5) | * |
| 35–39 | 222 | 11,355 | 2.0 | 385 | 11,753 | 3.3 | 1.7 | (1.4, 2.0) | * |
| 40–44 | 113 | 6,166 | 1.8 | 285 | 8,000 | 3.6 | 2.0 | (1.6, 2.5) | * |
| 45–49 | 65 | 3,688 | 1.8 | 159 | 5,554 | 2.9 | 1.6 | (1.2, 2.2) | 0.001 |
| 50–54 | 53 | 3,019 | 1.8 | 103 | 4,051 | 2.5 | 1.5 | (1.0, 2.0) | 0.03 |
| 55–59 | 30 | 2,425 | 1.2 | 62 | 2,990 | 2.1 | 1.7 | (1.1, 2.6) | 0.02 |
| 60 + | 50 | 3,349 | 1.5 | 107 | 4,609 | 2.3 | 1.6 | (1.1, 2.2) | 0.01 |

Data obtained from the National Hospital Morbidity Database for the period 1 July 2000 to 30 June 2020. PTH = post-tonsillectomy haemorrhage; CI = confidence interval.

*$P < 0.0001$.

**Table 3. Comparison of age-specific post-tonsillectomy haemorrhage rates between the adult and paediatric age groups between the male and female genders.**

| Gender | Age group (years) | PTH | Tonsillectomy procedures | PTH rate (%) | Odds ratio | 95% CI | P value |
|--------|-------------------|-----|--------------------------|--------------|------------|--------|---------|
| **Male** | Paediatric (1–14) | 3,312 | 333,332 | 1.0% | | Ref | |
| | Adult (15 +) | 5,154 | 124,604 | 4.1% | 4.3 | (4.1, 4.9) | * |
| **Female** | Paediatric (1–14) | 2,862 | 286,220 | 1.0% | 1.0 | (1.0, 1.1) | 0.80 |
| | Adult (15 +) | 4,057 | 196,995 | 2.1% | 2.1 | (2.0, 2.2) | * |

Data obtained from the National Hospital Morbidity Database for the period 1 July 2000 to 30 June 2020. PTH = post-tonsillectomy haemorrhage; CI = confidence interval.

*$P < 0.0001$.

increase since 2007–08. From 2015, there has been twice as many tonsillectomy procedures performed with adenoidectomy than without adenoidectomy. The pattern of increase in adenotonsillectomy directly matches the increase in the age-specific incidence of tonsillectomy procedures for paediatric patients aged 1 to 9 years as shown in Fig 4B. Fig 4C shows that the annual percentage of day-stay tonsillectomy procedures has steadily risen during the study period.

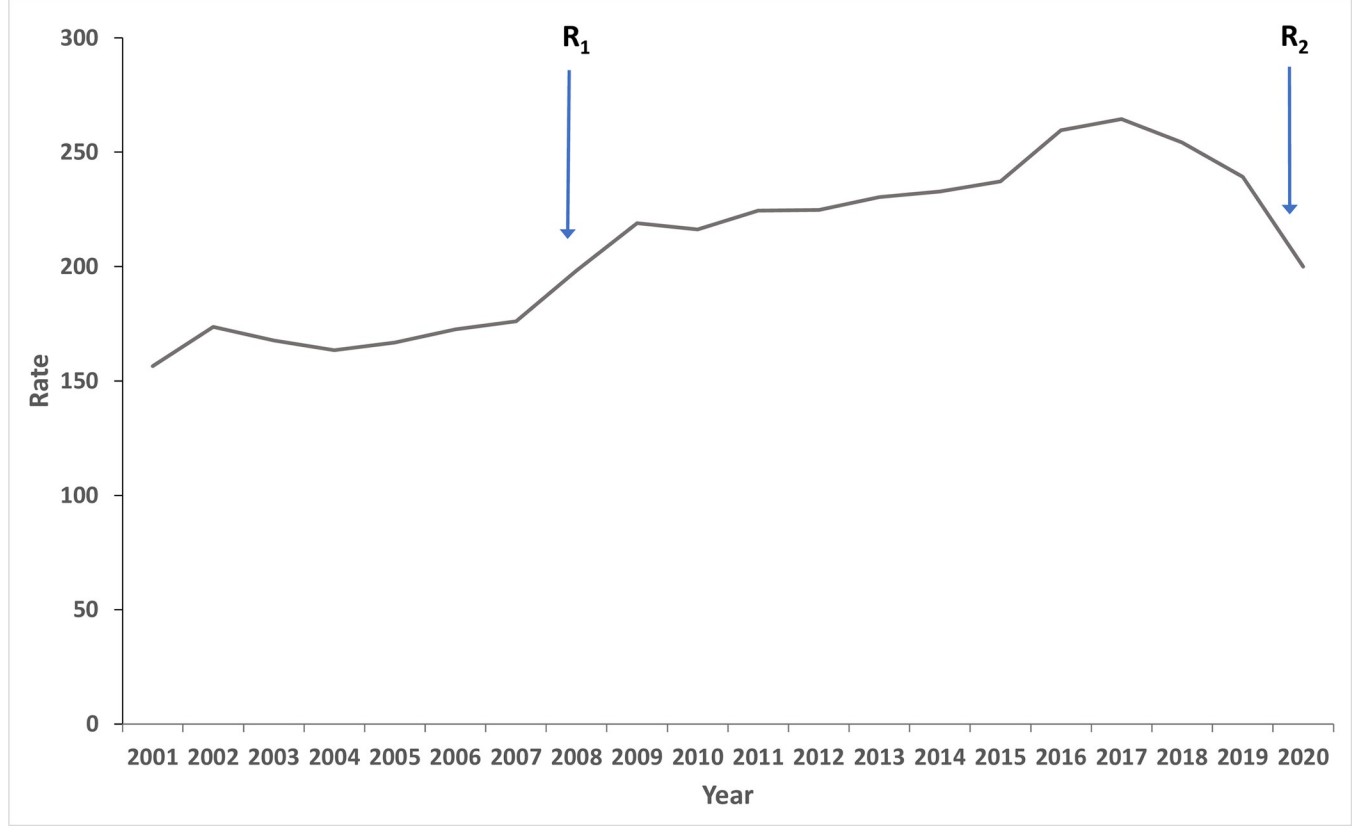

**Fig 3. Annual incidence of tonsillectomy procedures, 2000–01 to 2019–20.** Rate expressed in procedures per 100,000 person-years. $R_1$ arrow marks the release of the Guideline Indications for Tonsillectomy and Adenotonsillectomy released by the Royal Australasian College of Physicians and The Australian Society of Otolaryngology Head and Neck Surgery in July 2008. $R_2$ arrow marks the introduction of Australian Government's restrictions on elective surgery to free hospital resources to during the COVID-19 pandemic. Data obtained from the National Hospital Morbidity Database for the period 1 July 2000 to 30 June 2020.

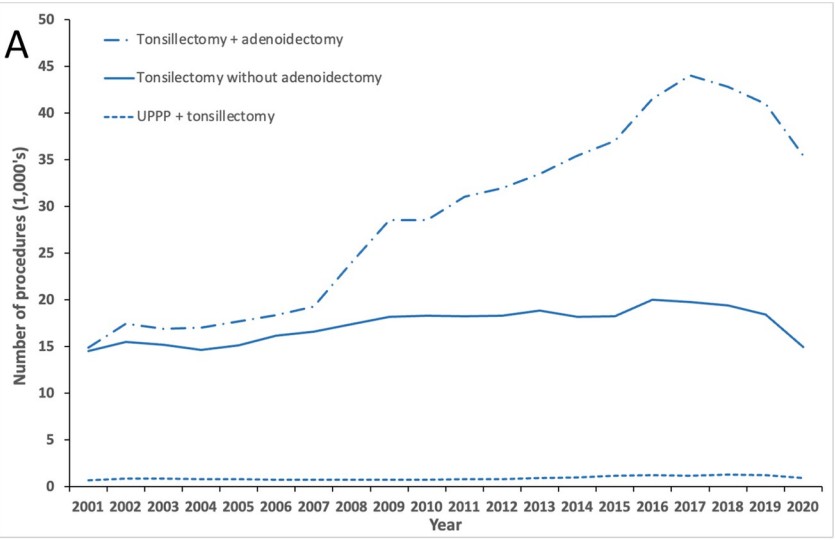

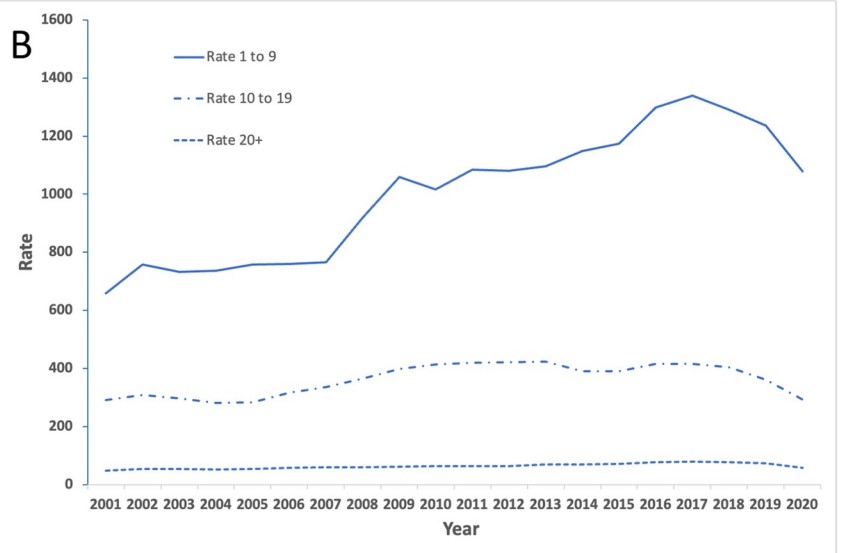

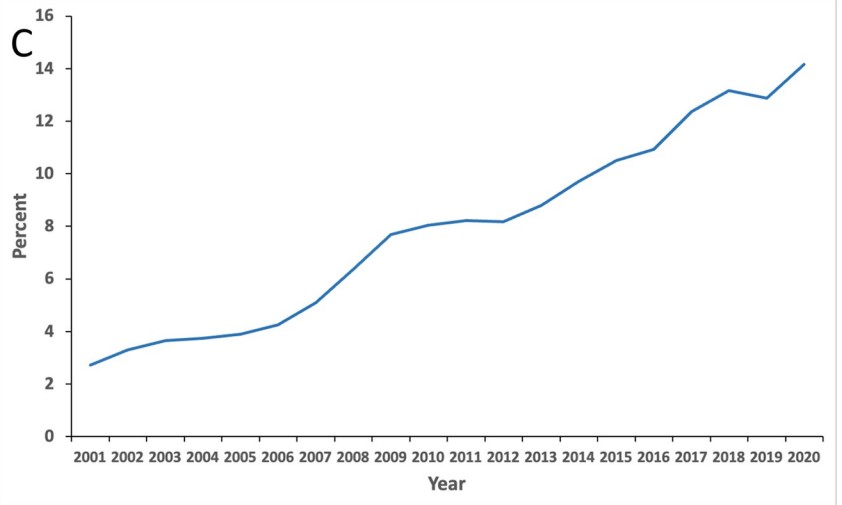

**Fig 4. Tonsillectomy procedure time-trends in Australia, 2000–01 to 2019–20.** Data obtained from the National Hospital Morbidity Database. A. Time-trend by tonsillectomy procedure type. Y axis displays number of procedures. Abbreviation UPPP; Uvulopharyngopalatoplasty. B. Time-trend by age group-specific incidence rates. Y axis displays rate per 100,000 person-years. C. Time-trend of percentage proportion tonsillectomy procedures performed as ambulatory day stay procedures.

**Age-specific incidence of tonsillectomy by gender.** The plots of age-specific incidence rates of tonsillectomy procedures for each gender are shown in Fig 5. Each line series represents the age-specific incidence rate averaged over a 5-year period to illustrate the changes in age-specific incidence rates over time. The age-specific incidence rate for females showed a clear bimodal pattern with a peak incidence between the 1 to 9 years and the 15 to 19 years age groups (Fig 5A). For males, a tapering of the age-specific rate for the 15 to 19 years age group was observed (Fig 5B). Comparison between the two figures shows that the incidence rate of tonsillectomy in males was consistently higher than that of females for the ages from 1 to 9 years. This relationship is reversed for the population aged between 10 to 24 years with the incidence rate for females exceeding males.

Age-specific incidence rate averaged over 5-year periods from 2000–01 to 2019–20 for females (A) and males (B). Rate expressed in procedures per 100,000 person-years. Data obtained from the National Hospital Morbidity Database.

## Discussion

Using data from the NHMD, we have conducted a national population-based analysis on the rates of PTH in tonsillectomy in Australia. Our study has found that over the past 20 years, the rate of PTH has essentially remained unchanged whilst the incidence of tonsillectomy procedures has substantially increased. This increase in tonsillectomy procedures has been predominantly driven by the increase incidence of paediatric tonsillectomies.

### Reference PTH rates

PTH rates that are widely considered as benchmarks rates had been derived either from national database studies [14–16, 19–24], large prospective cohort studies [25, 26] or national surgery quality registers, conducted in various countries [27, 28]. For comparative purposes, only studies that have reported PTH rates defined by severe haemorrhage episodes that required the return to theatre are listed in Table 4.

The foremost study that provided an account on the PTH rates on a nation-wide scale was the National Prospective Tonsillectomy Audit (NPTA) conducted in the UK (England and North Ireland) between 2003 and 2004 which consisted of 33,921 patients [25]. The NPTA study relied on information supplied by participating hospital units and voluntary consenting patients. One of the main findings of the NPTA was that hot surgical techniques that involve the use electrocautery, bipolar diathermy, Coblation or other thermal welding surgical were associated with a statistically higher risk of PTH than cold steel techniques. This finding resulted in the release of the interim national guidance in the UK on tonsillectomy by the British Association of Otorhinolaryngology Head and Neck Surgery (BAO-HNS) that cautioned the use of hot techniques for tonsillectomy. The study found an overall return to theatre rate of 1.0% and 0.9% before and after the release of the National guidance respectively. The NPTA investigators acknowledged in their report that their rates of PTH requiring the return to theatre were in fact under-estimated by up to fifty percent based on their comparison with the analysis of national hospital data from the UK's Hospital Episodes Statistics (HES) database. Possible causes include reduced data quality due to the potential of bias reporting from the

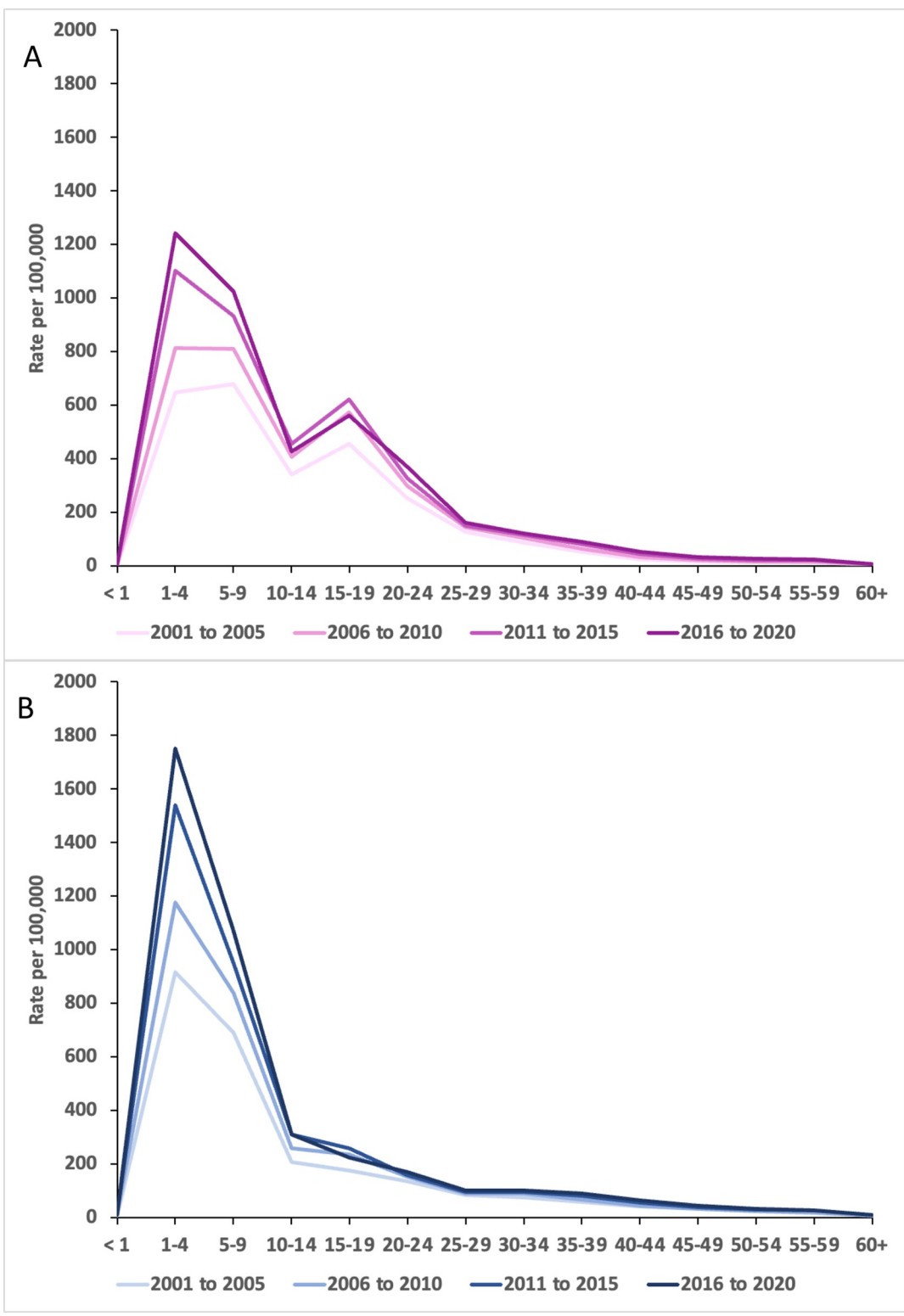

**Fig 5. Age-specific incidence of tonsillectomy procedures for males and females, 2000–01 to 2019–20.**

**Table 4. Summary of population studies in the literature with reported post-tonsillectomy haemorrhage rates defined by rates of return to theatre.**

| Author | Country | Study period | Study type | PTH rate (defined by return to theatre) |
|---|---|---|---|---|
| Tomkinson *et al.* 2005 [16] | UK (Wales) | 1999 to 2004 | Cross-sectional database analysis | 1.2–2.2% |
| Lowe *et al.* 2007 [25] | UK (England and Northern Ireland) | 2003 to 2004 | Prospective audit (NPTA) | 0.9% |
| Tomkinson *et al.* 2011 [15] | UK (Wales) | 2003 to 2008 | Cross-sectional database analysis | 1.5% |
| Sarny *et al.* 2011 [26] | Austria | 2009 to 2010 | Prospective multicentre cohort | 4.6% |
| Söderman *et al.* 2015 [27] | Sweden | 2009 to 2013 | Cross-sectional, national register (NTSRS) | 2.7% (tonsillectomy without adenoidectomy cases only) |
| Mueller *et al.* 2015 [19] | Germany (Thuringlia) | 2012 | Cross-sectional database analysis | 6.0% |
| Hallenstål *et al.* 2017 [28] | Sweden | 2013 to 2015 | Cross-sectional, national register (NTSRS) | 1.4% |
| Østvoll *et al.* 2018 [14] | Sweden | 1987 to 2013 | Longitudinal cross-sectional database analysis | 0.8% |
| Hsueh *et al.* 2018 [20] | Taiwan | 1997 to 2012 | Cross-sectional database analysis | 0.3% for patients < 18 years of age |
| Hsueh *et al.* 2019 [21] | Taiwan | 1997 to 2012 | Cross-sectional database analysis | 1% for patients > 20 years of age |
| Windfuhr and Chen 2019 [22] | Germany | 2005 to 2017 | Longitudinal cross-sectional database analysis | 5.4%, female; 7.6%, male |
| Milner *et al.* 2021 [23] | UK (Scotland) | 1998–2002 and 2013–2017 | Cross-sectional database analysis | 2.3%, 1998–2002; 3.1%, 2013–2017 |
| Keltie *et al.* 2021 [24] | UK (England) | 2008 to 2019 | Retrospective observational cohort database analysis | 1.2% for patients ≤ 16 years of age (0.4% and 0.8% for primary and secondary PTH, respectively) |
| This study | Australia | 2001 to 2020 | Longitudinal cross-sectional database analysis | 1.6% |

PTH = post-tonsillectomy haemorrhage, NPTA = National Prospective Tonsillectomy Audit, NTSRS = National Tonsil Surgery Register in Sweden

incomplete capturing of PTH episodes for the NPTA study. After taking the under-estimation into account, these guidance rates in the UK are comparable to the overall PTH rate we have calculated using our Australian data.

There are nuances in care practices by individual clinicians and hospital units between different countries, such as differences in the philosophies and processes of care surrounding tonsillectomy as well as the management of PTH. For example, the PTH rates as measured by return to theatre reported by studies conducted in Germany and Austria are some of the highest in the literature [22, 26]. The higher rates may reflect a more conservative approach to managing risks following tonsillectomy, which is demonstrated by their avoidance of outpatient or ambulatory tonsillectomies and their routine practice of several nights of hospital stays following tonsillectomy procedures [26, 29]. The higher rates in these countries may therefore be explained by a lower clinical threshold for re-operations due to PTH, rather than by reason of significant differences in the risk profile of their patient population or other factors such as choice of surgical techniques.

## Time-trend of PTH rate in Australia

This study also revealed that despite a stable annual rate of PTH for all ages over the 20-year period, there was a small but statistically significant trend increase in the incidence of PTH among adult patients.

We have used the non-parametric Spearman's rank correlation test in our study to avoid the violation of statistical assumptions that are required for the use of higher-power statistical

analysis models. Firstly, PTH rates are discrete epidemiological events that do not follow a normal distribution [30]. Secondly, in time-trends analysis, the consecutive time periods (year) are the predictor variable such that the PTH rate of one year may have correlations to the PTH rate in a following year. These correlations between consecutive years would violate the statistical assumption of independence that are required for the use of regression model for time-trend analyses [31].

This time-trend finding has real-world implications on patient outcomes concerning tonsillectomies. The statistically significant increase in the PTH rate for adults can be interpreted as an indicator of a decreased quality of care associated with adult tonsillectomy procedures. For example, the difference in PTH rate for adults between 2000–01 and 2019–20 was 0.6% which equates to 94 adult patients (0.6% x 15,672 adult tonsillectomies) encountering a return to theatre for PTH based on the number of adult tonsillectomies performed in 2019–20.

In contrast, an increasing trend in the rate of PTH in paediatric tonsillectomy was not detected by the time-trend analysis. One possible reason for the stability in the annual paediatric PTH rate may be due to an offset of any increasing trend in PTH rates by an increased adoption of tonsillotomy procedures over time. Tonsillotomy or partial intracapsular tonsillectomy is an alternative surgical procedure to conventional tonsillectomy for the treatment of paediatric obstructive sleep disordered breathing [32]. It has been shown to improve patient outcomes in terms of reducing post-operative pain [33], as well as having a reduced rate of post-operative haemorrhage compared to tonsillectomy [28, 33]. In some countries such as Sweden, the number of paediatric tonsillotomy procedures has increased significantly over time and currently outnumbers tonsillectomy procedures [34].

Unfortunately, we are unable to evaluate or estimate the influence of tonsillotomy procedures on paediatric rates of PTH as our study data does not differentiate tonsillotomy from tonsillectomy procedures. The incidence of tonsillotomy procedures in Australia is currently unknown.

There are other potential factors that may influence the national trend in PTH rates for both adult and paediatric tonsillectomies over time. Since the initial finding by the NPTA study, surgical techniques have been a subject of much investigation. The available evidence that supports the use of cold techniques (both for dissection and for haemostasis) over hot techniques to reduce PTH rates has been supported by other large perspective cohort studies, quality registers and large population studies [15, 26, 27, 35]. A study published in 2020 using data from the National Tonsil Surgery Register in Sweden (NTSRS) suggested that the recognition of the increased risk of PTH with the use of hot techniques has resulted in the subsequent stabilising of the national PTH rate in Sweden [35]. Furthermore, the association of cold techniques with lower rates of PTH has also been supported by the outcomes of quality improvement projects conducted in Sweden that specifically promoted the increased practice of cold techniques [36]. The increasing PTH rates in adult tonsillectomy in Australia may be explained by the preference in surgical techniques. In Australia, a survey study conducted in 2005 found that the hot technique of monopolar diathermy has been the preferred technique for tonsillectomy [37]. It is unknown how this practice has changed since the study, but the use of the other hot technique Coblation (also known as radiofrequency ablation) has been increasingly mentioned in Australian studies [10, 38, 39]. Whilst advocates of the use of Coblation offers less post-operative pain and faster return to normal diet following surgery, a systematic review has showed there is currently a lack of quality evidence to support these claims [40].

Another explanation for Australia's PTH rates for adults may be a potential change in the risk-profile of PTH in patients over time. This may occur as a result of clinicians becoming more stringent with the indication criteria for tonsillectomy, as suggested by Milner et al. as a conceivable reason for an increase in national rates of PTH over time as observed in Scotland

[23]. The joint position paper released in July 2008 by the Royal Australasian College of Physicians and The Australian Society of Otolaryngology Head and Neck Surgery provided a national guideline on the indications for tonsillectomy and adenotonsillectomy [41]. The implementation of this guideline corresponded with a subsequent marked increase in the incidence of paediatric tonsillectomy for the treatment of paediatric obstructive sleep apnoea, which suggests that the guideline was widely adopted by the Australian medical community. The guideline may also likely to have triggered more clinicians conforming with the criteria outlined by Paradise et al. for the performance of tonsillectomy for the treatment of recurrent acute tonsillitis [42]. This may have led to a higher proportion of patients meeting the stricter definition of recurrent or chronic tonsillitis, which in turn increases the overall risk and rate of PTH or the entire tonsillectomy patient cohort.

## Association of PTH with age and gender

This study extends the insight into the age-gender risk relationship of PTH reported in other large population studies [15, 16, 22]. Our study results show that the risk of PTH increased from age of one and plateaued beyond the age of 14 in females whilst the risk continued to increase sharply until the ages of 20 to 24 years for males. This indicates that increasing age is directly associated with the increased risk of PTH, with male patients in the 20 to 24 years age group having the highest risk of PTH. Although our datasets did not include information on the underlying indications for each tonsillectomy procedure, the higher rates of PTH in males were likely independent of the known association of infectious indications with higher rates of PTH. This is supported by studies that have shown that women are more likely than men to require tonsillectomy as treatment for chronic tonsillitis [43, 44]. In addition, our results show that for persons aged 15 to 24 years, more tonsillectomy procedures were performed in females than in males despite males having two to two and a half times the risk of PTH than females.

Tomkinson et al. [16] have suggested a possible association between the increased risk of PTH and puberty. This relationship is supported by our results and in particular the relationship between the risk of PTH and male puberty, suggesting male sex hormone testosterone may be associated with the risk of PTH. This link between PTH and testosterone may be due to the effects of circulating testosterone levels on the modulation of oropharyngeal tissue healing and mucosal re-epithelialisation that follow tonsillectomy. Experiments on oral mucosa in human subjects have shown that lower circulating testosterone levels were related to the faster healing of oral mucosal wounds in both young men and women [45]. This study also showed a significant difference in both the rate and temporal pattern of mucosal healing between younger and older men but not between younger and older women [45].

Using this age and gender relationship of PTH, we were able to analyse the odds of PTH based on age and gender. We found that in general, adult males (aged 15 years and over) had 4 times the odds of bleeding compared to persons in the paediatric age group (aged 1 to 14 years) and twice the odds of bleeding compared to adult females.

## Epidemiology of tonsillectomy in Australia

The observed increase in tonsillectomy procedures in Australia over time was shown to be mainly due to the increased incidence of paediatric adenotonsillectomy. Although our data does not contain information on the indications for each tonsillectomy procedure, this finding most likely reflects the growing recognition within the Australian medical community during the study period of the morbidity of obstructive sleep disordered breathing and obstructive sleep apnoea in children [41, 46].

Our results also showed that the number of tonsillectomy procedures in Australia plateaued in 2016–17 followed by a slight downward trend. The reason behind this is unclear, but the notable decrease in the incidence rate in 2019–2020 is most likely a result of the restrictions and suspensions on elective surgery that were initiated by the Australian government in response to the COVID-19 pandemic [47].

The tonsillectomy procedure count used in our study included tonsillectomies that were performed as part of UPPP surgery for obstructive sleep disordered breathing. Tonsillectomy with UPPP procedures were included in our study to provide a more complete representation of the haemorrhage rates following all forms of tonsillectomy procedures. The results show that tonsillectomy with UPPP represented only 2% of the total tonsillectomy procedures and therefore their inclusion does not statistically affect the estimation of our PTH rate calculations. Patients who undergo tonsillectomy with UPPP should be at no lesser or greater risk of developing PTH but tonsillectomy involving palate procedures are often excluded from studies examining PTH rates [48], possibly due to concern of palate procedures affecting the risk of PTH. There is a paucity of studies examining the PTH rates between UPPP alone and UPPP with tonsillectomy. However, data from two studies that allowed the comparison of PTH rates between tonsillectomy with or without UPPP showed the rate of PTH between these two procedures to be comparable and were not statistically different [49, 50].

Another finding from our study is a steady rise in the number of day-stay tonsillectomy cases in Australia over the last 20 years. Tonsillectomy performed as day-stay or ambulatory surgery in Australia has risen continuously from 2.7% in 2000–01 to 14.2% in 2019–20. Day-stay tonsillectomy, in itself, has not been found to be a risk factor for tonsillectomy and a Swedish study using data from the NTSRS showed that patients who had met the criteria for day-stay surgery have a lower risk of PTH [51]. A meta-analysis of 16 studies have also suggested overnight tonsillectomy conferred minimal benefit for patients in terms of PTH, with an estimate of only 1 of 14 episodes of primary haemorrhages noted to occur between 8 and 24 hours of surgery [52]. With the growing recognition of the safety of day-stay tonsillectomy, the practice of day-stay tonsillectomy has been increasingly adopted by some countries. In Sweden, day-stay has over time become the preferred care type for tonsillectomy [14]. Similarly, day-stay tonsillectomy, and in particular paediatric tonsillectomy, has also increased in the UK. Since 2008, an average of 46% of paediatric tonsillectomies in the UK were day-stay cases [24]. Conversely, in countries such as Germany and Taiwan, while the trends in the practice of day-stay tonsillectomy is unclear, in-patient surgery has remained the preferred treatment care type for tonsillectomy cases [19, 21].

## Strengths and limitations

This study utilised national hospital data covering a period of 20-years from the 1 July 2000 to 30 June 2020. The data derived from the NHMD enabled a high degree of data coherency, transparency and consistency for estimating a PTH reference rate for the Australian population. Further, as the data captures all tonsillectomy practices irrespective of clinicians or institutions, the findings in this study are generalisable and comparable for use by hospital unit and clinicians throughout Australia. Given that the source of our data is from the public domain, our results can be re-validated and continuously updated over time, without the need for additional resources and costs involved in data collection.

Nonetheless, there are known limitations in the use of national administrative health databases for epidemiological research including concerns over data completeness and data accuracy [53]. Under the provisions of the NHIA, health authorities of each State and Territory are responsible for the quality and completeness of the data they contribute to the NHMD.

According to the AIHW Data Quality Statement, in order to maintain and continually improve data quality, the AIHW undertakes extensive validations on receipt of data and cross checks the data with other datasets where possible [17]. Systematic coding and reporting errors can also affect data quality and lead to errors in the estimation of rates. However, these are likely to be rare occurrences and therefore should not significantly affect the results of our study.

Another limitation of our study is the inability to distinguish between primary and secondary PTH that are frequently reported in other PTH studies. However, we believe a return to theatre rate regardless of its primary or secondary nature provides a more universal indicator of the safety of tonsillectomy and is more suitable for use as a reference PTH rate. Further data-linkage projects using the existing NHMD datasets can help determine those return to theatre procedures that are involved in secondary PTH episodes. Another drawback of using administrative database for clinical research is the limited depth of clinical information available for analysis. An expanded NHMD datasets available through requests to the AIHW can be customised to include additional variables of interest. This would allow the analysis of other patient outcome parameters related to PTH such as the rates of blood transfusion and unplanned intensive care admissions as well as the mortality rates associated with PTH.

Additionally, the code that is used to identify return to theatre episodes for PTH (ACHI code 41797–00) is also used invariably for coding haemorrhage following adenoidectomy. However, haemorrhage following adenoidectomy that is sufficiently severe to require the return to theatre is relatively uncommon and has been estimated by a UK study to occur in 0.5% of adenoidectomy cases [54]. Adjusting for rates of adenoidectomy haemorrhage in adenotonsillectomy procedures would have the effect of correcting an over-estimation of the PTH reported in our study for the paediatric age group, and to a lesser degree, correcting an over-estimation of the overall PTH rate (i.e., for all ages) given adenoidectomy is an uncommon procedure among adults [55]. Further, patients that required multiple return to theatre for PTH were not identifiable from the NHMD datasets. There are no reports of the rate of multiple returns to theatre from Australian studies to estimate the size of this confounding factor. Correction for this factor would also lower our estimated rate of PTH.

## Conclusions

Using national hospital data, our study estimated the overall PTH rate, and more importantly, the age and gender-specific rates of PTH in Australia. These rates are defined by PTH episodes that required the need to return to theatre. Adult males (aged 15 years and over) had four times the risk of PTH compared to children (aged 1 to 14 years) and twice the risk compared to adult females. Our proposed national reference rates for PTH are generalisable for use by surgical audits and tonsillectomy quality assurance projects across Australia. The age and gender weighted PTH rates can be used during informed consent discussions and facilitate a more tailored explanation of PTH risks to individual patients. Our study also showed that the time-trend in the overall annual rates of PTH in Australia has been stable over the past two decades. Nevertheless, our findings indicate that the morbidity of tonsillectomy in Australia from a haemorrhage perspective has not improved over the study period as the annual rates of PTH for adult patients (aged over 15 years and over) have increased whilst the overall rates of PTH for all ages have not reduced over time. All clinicians should be mindful of the greater risk of PTH that is inherently present among male youths and young adults. Further large-multicentre studies or the adoption of a national tonsillectomy surgery register may help cross-validate our study findings and may provide continual improvements to the safety and quality of tonsillectomy procedures.

## Supporting information

**S1 Table. Annual rates of post-tonsillectomy haemorrhage in Australia, 2000–01 to 2019–20.** Data obtained from the National Hospital Morbidity Database for the period 1 July 2000 to 30 June 2020. PTH = post-tonsillectomy haemorrhage; CI = confidence interval; SD = standard deviation. Annual data are presented as absolute number of episodes of PTH and tonsillectomy procedures. PTH defined by re-operations for haemostasis. Annual PTH rates are calculated by dividing the number of PTH with the number of tonsillectomy procedures. The annual mean and standard deviation were calculated for the entire study period. (DOCX)

**S2 Table. Comparison of age-specific post-tonsillectomy haemorrhage rates between age groups for each gender.** Data obtained from the National Hospital Morbidity Database for the period 1 July 2000 to 30 June 2020. PTH = post-tonsillectomy haemorrhage; CI = confidence interval. $^*P < 0.0001$. (DOCX)

**S3 Table. Characteristics of tonsillectomy procedures in Australia.** Data obtained from the National Hospital Morbidity Database for the period 1 July 2000 to 30 June 2020. Data are presented as n (%). Total number of tonsillectomy procedures was calculated by the sum of tonsillectomy procedure codes. ACHI = The Australian Classification of Health Interventions; UPPP = Uvulopharyngopalatoplasty. (DOCX)

**S4 Table. Annual incidence of tonsillectomy procedures in Australia, 2000–01 to 2019–20.** Population data are the mid-year estimated resident population sourced from the Australian Bureau of Statistics (ABS). Absolute tonsillectomy procedure counts are obtained from the National Hospital Morbidity Database for the period 1 July 2000 to 30 June 2020. 95% confidence intervals for incidence rates are estimated using the Poisson regression model. (DOCX)

## Acknowledgments

We wish to acknowledge the Australian Institute of Health and Welfare for data from the National Hospital Morbidity Database.

## Author Contributions

**Conceptualization:** Jonathan C. Li.

**Data curation:** Jonathan C. Li.

**Formal analysis:** Jonathan C. Li, David Veivers.

**Investigation:** Jonathan C. Li, Martin Forer, David Veivers.

**Methodology:** Jonathan C. Li, David Veivers.

**Project administration:** Jonathan C. Li, Martin Forer, David Veivers.

**Resources:** Jonathan C. Li, Martin Forer.

**Software:** Jonathan C. Li.

**Supervision:** Jonathan C. Li, Martin Forer, David Veivers.

**Validation:** Jonathan C. Li, David Veivers.

**Visualization:** Jonathan C. Li, David Veivers.

**Writing – original draft:** Jonathan C. Li.

**Writing – review & editing:** Jonathan C. Li, Martin Forer, David Veivers.

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
