## [Decision Letter · Decision Letter 0]

30 Jun 2022

PONE-D-22-10343Reference rate for post-tonsillectomy haemorrhage in Australia - A 2000-2020 national hospital morbidity database analysisPLOS ONE

Dear Dr. Li,

Thank you for submitting your manuscript to PLOS ONE. After careful consideration, we feel that it has merit but does not fully meet PLOS ONE’s publication criteria as it currently stands. Therefore, we invite you to submit a revised version of the manuscript that addresses the points raised during the review process.

We look forward to receiving your revised manuscript.

Kind regards,

Sethu Thakachy Subha, M.S

Academic Editor

PLOS ONE

Journal Requirements:

Reviewers' comments:

Reviewer's Responses to Questions

**Comments to the Author**

1. Is the manuscript technically sound, and do the data support the conclusions?

Reviewer #1: Yes

2. Has the statistical analysis been performed appropriately and rigorously? 

Reviewer #1: Yes

3. Have the authors made all data underlying the findings in their manuscript fully available?

Reviewer #1: Yes

4. Is the manuscript presented in an intelligible fashion and written in standard English?

Reviewer #1: Yes

5. Review Comments to the Author

Reviewer #1: I enjoyed reading this paper. Tonsillectomy is so common, and bleeding such a common and serious complication, that it is perhaps surprising that so few national datasets have been published like this (the authors may wish to check out a new paper from the UK which probably wasn't available when they wrote the manuscript: https://doi.org/10.1111/coa.13707).

This paper is a useful addition to the literature and I would be keen to see it published. Is there any comment the authors could make on data completeness in the NHMD (ie the proportion of hospital episodes for which a complete and valid data return is made for the NHMD)? Also, could the authors comment on the possibility of miscoding (eg post-tonsillectomy haemorrhage being coded under some other "arrest of haemorrhage" code)?

I find Table 1 misleading and unhelpful. I don't think anyone is interested in what proportion of all the bleeds occur in a particular age group because of course the number of tonsillectomies done varies so much between age groups that this proportion is meaningless. Table 1 would be more useful if it presented the haemorrhage rate for each age group. The same is true in the text: "the age group that had the highest proportion of PTH being the 15 to 19 years age group (19.2%) followed by the 5 to 9 years age group (17.1%)". More bleeds occur at these ages because more tonsillectomies are done at these ages. The reader is easily misled by this into thinking these are the ages with the highest bleed rates which is not the case. The actual bleed rates seem to rise smoothly through adolescence and early adulthood, so the highest rates are between roughly 15 and 35 years of age.

I am not sure Table 2 adds much either. I would be happy just to see the statement in the text that there is no discernible time trend in the overall numbers and then move on. Figure 1a gives us the information anyway. Table 3 is much more interesting, with the adult and paediatric data presented separately, together with Figure 1b.

Figure 2 and Table 4 are very interesting, with the difference in bleed rates between males and females being quite marked. In fact the biggest difference occurs in the young adult years which happens to be when the highest overall bleed rates occur, and the male bleeds largely account for the increase in the overall bleed rates at this age. Personally, I would combine this information with a table of overall bleed rates in table 1 to tell this story more clearly. I don't think Table 5 adds much: do we really need this in addition to tables 4 and 6?

Discussion and references are comprehensive. So my only real gripes are about the sheer volume of information presented in tables which is a bit overwhelming and serves to obscure the important messages in the paper: trim these a little and the more streamlined paper will be easier to follow.

6. PLOS authors have the option to publish the peer review history of their article (what does this mean?). If published, this will include your full peer review and any attached files.

Reviewer #1: **Yes: **Haytham Kubba

---

## [Author Response · Author response to Decision Letter 0]

14 Jul 2022

Dear Dr Sethu Thakachy Subha,

Many thanks for reviewing our manuscript. Please find enclosed our revised manuscript (Marked-up and clean copies) as well as our 'Response to Reviewers' letter.

Kind regards

Jonathan Li

---

## [Decision Letter · Decision Letter 1]

8 Aug 2022

Reference rate for post-tonsillectomy haemorrhage in Australia - A 2000-2020 national hospital morbidity database analysis

PONE-D-22-10343R1

Dear Dr. Li,

We’re pleased to inform you that your manuscript has been judged scientifically suitable for publication and will be formally accepted for publication once it meets all outstanding technical requirements.

Kind regards,

Sethu Thakachy Subha, M.S

Academic Editor

PLOS ONE

Additional Editor Comments (optional):

Reviewers' comments:

Reviewer's Responses to Questions

**Comments to the Author**

1. If the authors have adequately addressed your comments raised in a previous round of review and you feel that this manuscript is now acceptable for publication, you may indicate that here to bypass the “Comments to the Author” section, enter your conflict of interest statement in the “Confidential to Editor” section, and submit your "Accept" recommendation.

Reviewer #1: All comments have been addressed

2. Is the manuscript technically sound, and do the data support the conclusions?

Reviewer #1: Yes

3. Has the statistical analysis been performed appropriately and rigorously? 

Reviewer #1: Yes

4. Have the authors made all data underlying the findings in their manuscript fully available?

Reviewer #1: Yes

5. Is the manuscript presented in an intelligible fashion and written in standard English?

Reviewer #1: Yes

6. Review Comments to the Author

Reviewer #1: Thank you for addressing the comments. This is a good paper and I would support its publication in PLOS ONE.

7. PLOS authors have the option to publish the peer review history of their article (what does this mean?). If published, this will include your full peer review and any attached files.

Reviewer #1: **Yes: **Haytham Kubba

---

## [Editor Report · Acceptance letter]

10 Aug 2022

PONE-D-22-10343R1 

Reference rate for post-tonsillectomy haemorrhage in Australia - A 2000-2020 national hospital morbidity database analysis 

Dear Dr. Li:

I'm pleased to inform you that your manuscript has been deemed suitable for publication in PLOS ONE. Congratulations! Your manuscript is now with our production department. 

Kind regards, 

on behalf of

Dr. Sethu Thakachy Subha 

Academic Editor

PLOS ONE